# An Improved Model Based Detection of Urban Impervious Surfaces Using Multiple Features Extracted from ROSIS-3 Hyperspectral Images

**Yuliang Wang** [1,2] **, Huiyi Su** [1] **and Mingshi Li** [1,3,*]

1   College of Forestry, Nanjing Forestry University, Nanjing 210037, China; ylw@chzu.edu.cn (Y.W.); suhuiyi_agirl@163.com (H.S.)
2   School of Computer and Information Engineering, Chuzhou University, Chuzhou 239000, China
3   Co-Innovation Center for Sustainable Forestry in Southern China, Nanjing Forestry University, Nanjing 210037, China
*   Correspondence: nfulms@njfu.edu.cn or ylw@chzu.edu.cn; Tel.: +86-25-8542-7327

**Abstract:** Hyperspectral images (HSIs) provide unique capabilities for urban impervious surfaces (UIS) extraction. This paper proposes a multi-feature extraction model (MFEM) for UIS detection from HSIs. The model is based on a nonlinear dimensionality reduction technique, t-distributed stochastic neighbor embedding (t-SNE), and the deep learning method convolutional deep belief networks (CDBNs). We improved the two methods to create a novel MFEM consisting of improved t-SNE, deep compression CDBNs (d-CDBNs), and a logistic regression classifier. The improved t-SNE method provides dimensionality reduction and spectral feature extraction from the original HSIs and the d-CDBNs algorithm extracts spatial feature and edges using the reduced dimensional datasets. Finally, the extracted features are combined into multi-feature for the impervious surface detection using the logistic regression classifier. After comparing with the commonly used methods, the current experimental results demonstrate that the proposed MFEM model provides better performance for UIS extraction and detection from HSIs.

**Keywords:** urban impervious surface; multi-feature extraction; dimensionality reduction; deep learning; hyperspectral images

## 1. Introduction

Information on the distribution and sprawl of urban impervious surfaces (UIS) is crucial for urban planners and government decision-making. Currently, this information is widely used for urban land planning, urban heat island monitoring, illegal construction detection and environmental inspection [1,2]. Accurate UIS detection highly depends on the sensor type, quality, resolution, spectral and spatial information, and other features of remotely sensed data. Hyperspectral images (HSIs) generally provide moderate or relatively high spatial resolution and hundreds of spectral bands ranging from the visible to shortwave regions [3–5]. These data provide sufficient detail to delineate landscapes and have advantages that improve the extraction accuracy of UIS.

However, UIS extraction from HSIs has several challenges: (1) The data have high spatial and spectral variability [6]. HSIs with high spectral resolution provide fine spectral detail and rich information on land cover types but this easily leads to confusion in spatial domain, and the spectral information is relatively complex. (2) Unlabeled data affect the detection accuracy and the training sample and test sample selection [7]. (3) Dimensionality reduction (DR) is especially challenging [8] because HSIs have high dimensionality due to hundreds of spectral bands, resulting in the Hughes phenomenon [3]. DR is required prior to feature extraction and classification.

To address these problems, many methods have been proposed and developed for UIS extraction in recent decades [9]. Early on, machine learning methods were considered popular extraction methods for HSIs. Common methods include decision trees [10], logistic regression (LR) [11], minimum distance [12], maximum likelihood [13], k-nearest neighbor methods [14], and random forest [15]. A support vector machine (SVM) is a classical nonlinear classification algorithm, in which the number and type of samples are determined manually [16,17]. However, these extraction methods consist of a single-layer feature extraction model and deeper features cannot be extracted. Commonly, the spectral bands are selected from hundreds of bands and a transformation of the spectral matrix is performed, which results in information loss [10]. In addition, these methods also have to consider DR and labeling of the samples. Therefore, a DR method for the full spectrum and deep-feature extraction methods are required.

The goal of DR is to reduce the high-dimensional (HD) data to a low-dimensional (LD) subspace representation with intrinsic spatial-spectral features and attribute values. Generally, spectral-based DR methods are classified into supervised and unsupervised methods [18]. Supervised DR methods use labeled samples in defined classes to identify land objects based on different features. These methods include linear discriminant analysis (LDA) [19], local discriminant embedding (LDE) [20], and local Fisher discriminant analysis (LFDA) [21]. Unsupervised DR methods provide LD classification data by using a transformation matrix without labeled samples. Principal component analysis (PCA) and t-distributed stochastic neighbor embedding (t-SNE) [22] are two representative unsupervised DR algorithms. In the PCA method, a linear transformation matrix is created and its sum of squared errors is minimized [23]. However, the method has poor performance for data with a subspace structure and is unsuitable for sparse matrices and large amounts of data. In comparison, the t-SNE method is considered a significantly better spectral-based DR algorithm and provides HD data visualization for HSIs [24]. In the t-SNE method, a nonlinear transformation matrix from the HD space to an LD representation (e.g., 2-D or 3-D) is created. A t-distributed model is created that minimizes the distance between data points following the one-degree-of-freedom distribution in the LD space. In the HD space, the data points follow a Gaussian distribution. However, the cost function of the t-SNE algorithm does not guarantee a global optimum due to the large-scale computation with few labeled samples and lack of algorithm predictability. Therefore, the optimization of the method and the integration of a regression model are required for unsupervised spectral feature extraction.

Moreover, the use of spatial features can improve the classification accuracy of land covers. This method reduces the confusion between land cover classes with similar spectral features and improves the classification performance, which has been demonstrated in previous studies [4,25]. Several spatial/spectral-based extraction models have been proposed in recent years [26,27]. These methods consider both spatial and spectral information for features extraction. However, the traditional spatial feature extraction depends on the selection of spatial training samples. These training samples correspond to specific land cover classes and empirical knowledge is required for image interpretation. Additionally, spatial attributes are variable and cannot represent all land cover types and this affects the classification accuracy [28].

Deep learning methods are considered more advanced machine learning approaches [29] and consist of high-level and multi-layer networks for the automatic extraction of spatial features. Moreover, many unlabeled samples are not a problem for deep learning methods. Common deep learning approaches include deep Boltzmann machines (DBMs) [30], stacked denoising auto-encoders (SDA) [31], convolutional neural networks (CNNs) [32], deep auto-encoders (DAE) [33], and deep belief networks (DBNs) (see Abbreviations) [27,34]. However, these algorithms require fixed-scale detection windows, which is unsuitable for detecting spatially and spectrally variable land cover objects in HSIs. To address this problem, convolutional DBNs (CDBNs) have been proposed [35]. The method represents an unsupervised learning approach for a two-dimensional (2D) image structure and is based on DBNs. It is a hierarchical generative model with full-sized image transformation and uses probabilistic max-pooling (PMP) in a multi-layer architecture for high-level representations and

multi-layer edge detection. However, there is room for improvement regarding the shared weights of the layer connection.

In this paper, we propose a multi-feature extraction model (MFEM) for UIS detection from HSIs. The model combines the t-SNE-based approach and the CDBNs-based framework for HSIs interpretation for spectral, spatial, and edge feature extraction. In the MFEM, we propose an improved t-SNE method. Comparing to the standard t-SNE, the improved version has the advantages of reduced time complexity, improved similarity of the interclass data points, and better predictability. The method combines DR and spectral feature extraction. In addition, we improved the CDBNs-based algorithm, which we call deep compression CDBNs (d-CDBNs). The d-CDBNs method can shrink sharing weights of the energy function and reduces the redundancy of features when analyzing full-scale images. The d-CDBNs method combines the features of a multi-layer convolutional restricted Boltzmann machine (CRBM) and (PMP). The two functions provide unsupervised learning and training for full-sized 2-D images. The d-CDBNs method extracts spatial features and detects edges using a hierarchical generative framework. Finally, the spectral features, spatial features, and edge information are combined using a multi-feature extraction strategy based on an LR classifier. The MFEM model has better characteristics than other commonly used methods (Table 1). The main contributions of our work are as follows.

- **Development of the MFEM for UIS detection from HSIs.** The MFEM model has three main components, i.e., DR and spectral feature extraction, spatial feature and edge detection, and multi-feature classification. The model is an integration of the improved t-SNE model, the d-CDBNs, and the LR. Compared with commonly used methods, the model uses an unsupervised and nonlinear DR method, requires fewer labeled samples, has multi-layer feature networks, and a multi-feature cooperation extraction mechanism.
- **Improvement of the t-SNE method.** The improved t-SNE model has lower time complexity, improved similarity evaluation performance of the interclass data points, an embedded LR algorithm to determine the global and local optima, and better results. Compared with the standard t-SNE method, the improved method has a faster neighbor point search, better similarity detection performance, and a better prediction function.
- **Improvement of the CDBNs method.** The proposed d-CDBNs method markedly reduces the redundancy of the shared weights of the layer connection. Unlike the original CDBNs method, the improved method provides deep compression for the shared weights to reduce the data volume and the weight redundancy of the layer connection for the convolution operation.
- **Edge information extraction from HSIs.** The MFEM model detects edge information of landscapes using sparse regularization to reduce the confusion between UIS and other land cover classes with similar spectral information. The integration of the spatial and spectral features and the edge detection reduces the "salt and pepper" noise of the classification results.

**Table 1.** Comparison between our model and commonly used methods.

| Reference | Unsupervised | Extraction Layers | Edge Detection | Labeled Data | Multi-Feature Extraction | Nonlinear | Shared Weights |
|---|---|---|---|---|---|---|---|
| [19] | × | single | × | more | × | × | × |
| [20] | × | single | × | more | × | ○ | × |
| [21] | × | single | × | more | × | ○ | × |
| [22] | ○ | single | × | less | × | ○ | × |
| [23] | ○ | single | × | less | × | × | × |
| [31] | × | multiple | × | more | × | ○ | × |
| [32] | × | multiple | × | more | × | ○ | × |
| [33] | × | multiple | × | more | × | ○ | × |
| [34] | × | multiple | × | more | × | ○ | × |
| [35] | × | multiple | × | more | × | ○ | ○ |
| The proposed MFEM | d-CDBNs (○) | multiple | ○ | less | ○ | ○ | d-CDBNs (○) |

The rest of paper is organized as follows. In Section 2, the relationship between the t-SNE-based and the CDBNs-based architecture, as well as the improved methods and the MFEM framework are described. The experimental results of the analysis of two HSIs are presented in Section 3. In the last section, results are discussed and we accordingly conclude our work.

## 2. Materials and Methods

### 2.1. Dataset Description

Two popular hyperspectral datasets were selected to demonstrate the performance of the MFEM. One is the university scene and the other is the city center scene in Pavia in northern Italy (http://www.ehu.eus/ccwintco/index.php). The two datasets have universal representative characteristics that are suitable for the validation of the proposed model. The characteristics include the following: (1) The two datasets are HD datasets with hundreds of bands, covering the electromagnetic spectrum range from 0.43 to 0.86 μm. The images were acquired by the ROSIS-3 sensor with high spatial resolution (1.3 m). (2) The images cover different impervious surfaces with complex man-made objects. The university scene has sparse buildings and the city center scene has dense buildings. (3) The datasets include ground-truth information for nine land cover classes, representing a certain difficulty for classification. These characteristics allowed us to determine the performance of the improved method and obtain universal experiment results.

The university scene has a size of $610 \times 340$ pixels. Its 103 effective spectral bands were selected for excluding noisy bands. The dataset has nine classes: trees, meadows, bare soil, shadows, asphalt, bricks, bitumen, gravel, and metal sheets. The latter five classes represent impervious surfaces. The shadows represent pervious surfaces and occur mostly close to sparse buildings. The dataset is shown in Figure 1 and the list of classes is displayed in Table 2.

In the city center scene, similarly, we effectively selected 102 bands with a size of $1096 \times 715$ pixels. This dataset covers an area with dense urban buildings with complex spatial structure. The buildings cast many shadows and obscure other objects, resulting in difficulty in the object identification. The dataset has nine classes: trees, water, meadows, bare soil, shadows, asphalt, bitumen, bricks, and tiles. The latter four classes represent impervious surfaces. The reason that shadows are excluded from the impervious surface is explained in Section 4. The dataset is shown in Figure 2 and the details of the classes are listed in Table 3. In Tables 2 and 3, the reference land cover data are included. The ground truth image and impervious surface image were considered the reference data for the classification (Figures 1 and 2).

**Table 2.** The dataset display of land-cover classes and reference data in the university scene of the Pavia, Italy.

| Class Code | Class | No. of Reference Data | Reference Cover Property | |
|---|---|---|---|---|
| | | | Impervious (▲) | Pervious (△) |
| 1 | Tree | 1494 | - | △ |
| 2 | Meadow | 9436 | - | △ |
| 3 | Bare soil | 2489 | - | △ |
| 4 | Shadow | 586 | - | △ |
| 5 | Asphalt | 4931 | ▲ | - |
| 6 | Brick | 2787 | ▲ | - |
| 7 | Bitumen | 549 | ▲ | - |
| 8 | Gravel | 585 | ▲ | - |
| 9 | Metal sheet | 670 | ▲ | - |
| Total | | 23,527 | 5 | 4 |

**Table 3.** The dataset display of land-cover classes and reference data in the city center scene of the Pavia, Italy.

| Class Code | Class | No. of Reference Data | Reference Cover Property | |
|:---:|:---:|:---:|:---:|:---:|
| | | | Impervious (▲) | Pervious (△) |
| 1 | Tree | 1065 | - | △ |
| 2 | Water | 22,491 | - | △ |
| 3 | Meadow | 831 | - | △ |
| 4 | Bare soil | 959 | - | △ |
| 5 | Shadow | 657 | - | △ |
| 6 | Asphalt | 2339 | ▲ | - |
| 7 | Bitumen | 1753 | ▲ | - |
| 8 | Brick | 989 | ▲ | - |
| 9 | Tile | 16,813 | ▲ | - |
| Total | | 47,897 | 4 | 5 |

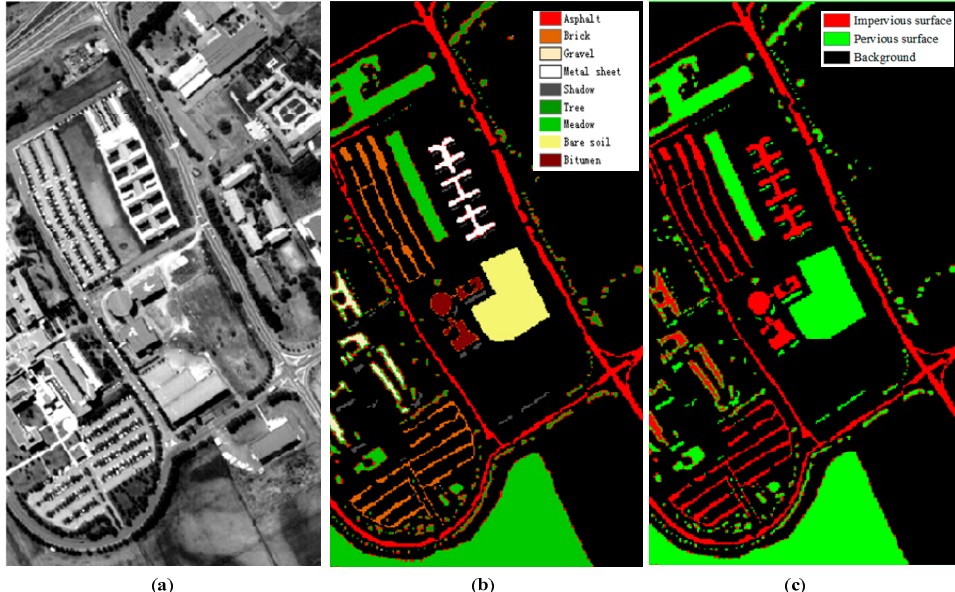

**Figure 1.** The university scene of the Pavia: (**a**) single band image; (**b**) reference of ground-truth image; and (**c**) reference of impervious surface image.

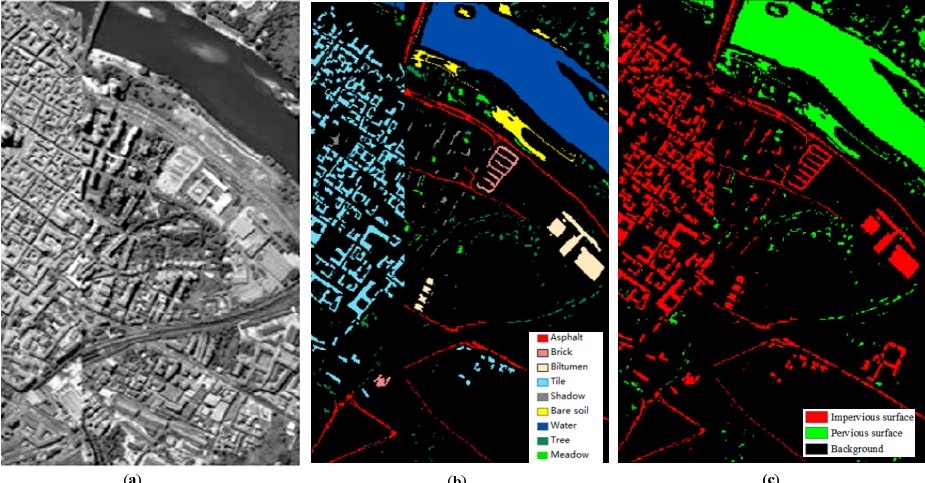

**Figure 2.** The city center scene of the Pavia: (**a**) single band image; (**b**) reference of ground-truth image; (**c**) reference of impervious surface image.

In the MFEM model, the improved t-SNE DR model and d-CDBNs method are combined into an effective multi-feature extraction strategy. The algorithmic theories of the t-SNE and CDBNs method constitute the foundation of the MFEM model. Therefore, in the following subsection, an overview of the t-SNE and CDBNs algorithms is described. Moreover, the improvements of the two methods for multi-feature extraction are described.

*2.2. t-SNE*

The t-SNE method is a nonlinear and unsupervised DR approach and is based on stochastic neighbor embedding (SNE) [36]. The t-SNE method alleviates the crowding problem of the heavier tails of the t-distribution, which takes the place of the Gaussian distribution in the LD map. The LD points are the result of embedding HD points with similarities and correspond to nearby and distant points, respectively. The t-SNE captures both the local and global structure of the HD input data for a multi-scale representation.

Commonly, nonlinear DR methods transform the HD input dataset $D_X = \{x_1, x_2, x_3 \cdots, x_N\}$ into a LD output dataset $D_Y = \{y_1, y_2, y_3 \cdots, y_N\}$ using a map function. The LD output dataset represents the HD input dataset as much as possible to preserve the significant features of the original dataset in the nonlinear manifold. A distance function $d_x(x_i, x_j)$ is used, which represents the distance between pairwise points, $x_i$, $x_j \in D_X$. The distance function uses the standard Euclidean distance in the t-SNE, $d_x(x_i, x_j) = ||x_i - x_j||$. For the DR, the joint probabilities $p_{ij}$ are defined to evaluate the pairwise similarity between the input data $x_i$ and $x_j$ by symmetrizing the conditional probabilities $p_{j|i}$ and $p_{i|j}$. The joint probabilities are defined as

$$p_{ij} = \frac{p_{j|i} + p_{i|j}}{2N}. \tag{1}$$

where $N$ is the number of rows in the input datasets $D_X$ and $p_{j|i}$ is the conditional probability of the similarity between the input data points $x_i$ and $x_j$ as follows:

$$p_{j|i} = \frac{exp\left(-d_X(x_i, \ x_j)^2/2\sigma_i^2\right)}{\sum_{k \neq i} exp\left(-d_X(x_i, \ x_k)^2/2\sigma_i^2\right)}, \quad pi|i = 0. \tag{2}$$

where $p_{j|i}$ is the conditional probability of $x_j$ as $x_i$ neighboring points, if the neighbors are collected according to the probability density of $x_j$ at a Gaussian kernel $x_i$. The conditional probability $p_{j|i}$ has a higher value of the adjacent data points $x_j$ than the separated data points $x_j$. However, the value of $p_{j|i}$ is dependent on reasonable values of the Gaussian variance, $\sigma_i$. A perplexity is determined for each input data point $x_i$ using a binary search algorithm with the aim that the perplexity of the conditional distribution $P_i$ represents the predefined perplexity of all input values of $x_i$. The perplexity function is defined as the number of neighboring points at a Gaussian center $x_i$ [37]. The perplexity function is defined as follows:

$$Perplexity(P_i) = 2^{H(P_i)}, \tag{3}$$

$$H(P_i) = -\sum_j p_{j|i} log_2\left(p_{j|i}\right). \tag{4}$$

where $H(P_i)$ denotes the Shannon entropy of the probability distribution $P_i$. If $P_i$ has a high entropy $H(P_i)$ value, the perplexity improves with the entropy and $\sigma_i$ has a higher configuration parameter, i.e., the number of nearby points will increase in the probability distribution. Generally, the empirical value of the perplexity is set in the range from 5 to 50.

Our goal is to learn and achieve $N_l$-dimensional $D_Y$ representing each point in the input dataset $D_X$ by stochastically embedding the points with a minimal Kullback–Leibler (*KL*) divergence. Where possible, the value of $N_l$ is usually set to 2 or 3. The t-SNE method uses a Student's t-distribution in

the LD space $D_Y$ to learn and optimize the Gaussian distribution in the HD space $D_X$ and computes the similarities $d_y$ between each pairwise points $d_y$, $d_y(y_i, y_j) = ||y_i - y_j||$. Correspondingly, the joint probabilities $q_{ij}$ are defined as follows:

$$q_{ij} = \frac{\left(1 + d_y(y_i, \ y_j)^2\right)^{-1}}{\sum_k \sum_{l \neq k} \left(1 + d_y(y_k, \ y_l)^2\right)^{-1}}, \ q_{ii} = 0. \tag{5}$$

where $q_{ij}$ is the embedding similarity between $y_i$ and $y_j$ in the $N_l$-dimensional $D_Y$. It is achieved by using a Student's t-distribution with a single degree of freedom. To minimize the *KL* divergences between the $p_{j|i}$ and $p_{i|j}$, the joint probability distribution represents the conditional probability distribution

$$C(D_Y) = KL(P||Q) = \sum_i \sum_j p_{ij} log \frac{p_{ij}}{q_{ij}}. \tag{6}$$

where $P$ is the HD input dataset and $Q$ is the $N_l$-dimensional output dataset. We assume $p_{ji} = 0$ and $q_{ji} = 0$. In the symmetric SNE, for $\forall \ i, j$, the joint probability $p_{ij} = p_{ji}$ and $q_{ij} = q_{ji}$. The HD map $p_{ij}$ and the $N_l$-dimensional map $q_{ij}$ are optimized as follows:

$$p_{ij} = \frac{exp\left(-d_x(x_i, \ x_j)^2/2\sigma^2\right)}{\sum_{k \neq l} exp\left(-d_x(x_k, \ x_l)^2/2\sigma^2\right)}, \qquad q_{ij} = \frac{exp\left(-d_y(y_i, \ y_j)^2\right)}{\sum_{k \neq l} exp\left(-d_y(y_k, \ y_l)^2\right)}. \tag{7}$$

In the symmetric SNE, a modified gradient descent algorithm to minimize the *KL* divergence is given by:

$$\frac{\partial C(D_Y)}{\partial y_i} = 4 \sum_{j \neq i} \left(p_{ij} - q_{ij}\right) q_{ij} \tau (y_i - y_j). \tag{8}$$

where $\tau$ is a normalization term that is defined as $\tau = \sum_{k \neq l} \left(1 + d_y(y_k, \ y_l)^2\right)^{-1}$.

The t-SNE DR architecture reduces the negative effects of the "crowding problem" due to the symmetry of the SNE and retains the local and global features of the original object. However, the classification accuracy and learning rate of the t-SNE require fine-tuning, as does the algorithm efficiency.

### 2.3. Spectral Feature Extraction with the Improved t-SNE

The new optimized model inherits the advantages of the traditional t-SNE method but provides improved DR performance and more accurate extraction of the spectral features. We propose an improved t-SNE model for reducing the algorithm time complexity, increasing the prediction performance, and improving the similarity. The similarity refers to the relationship between the HD joint probabilities $p_{ij}$ and the $N_l$-dimensional embedding probabilities $q_{ji}$. The optimizing strategy includes: (1) developing a faster search algorithm for detecting similar data points; (2) determining the similarity of the data points; and (3) embedding a LR model.

In the t-SNE, the similarities of the input dataset $D_X$ are considered a normalized Gaussian center $x_i$, if the neighboring points $x_j$ is a dissimilar data point. In this case, the joint probabilities $p_{ji}$ have a lower and negligible value. This allows performing a sparse approximation, and reducing the negative effects of dissimilar data points. We assume that $\lambda_i$ represents the adjacent neighboring datasets of input $x_i$ and the condition of the perplexity $p_{j|i}$ is redefined as follows:

$$p_{j|i} = \begin{cases} \frac{exp\left(-d(x_i, x_j)^2/2\sigma^2\right)}{\sum_{k \in \lambda_i} exp\left(-d(x_i, x_k)^2/2\sigma^2\right)}, & j \in \lambda_i \\ 0, & otherwise \end{cases} \tag{9}$$

Herein, the Gaussian variance $\sigma_i$ is set so that there is equivalence between the perplexity of the conditional distribution and the predefined perplexity. For a faster search of the neighboring points, an alternate algorithm with disordered dataset sorting, i.e., a hash table search algorithm, is used to replace the binary search algorithm. The hash table search time complexity for O (1) is superior to the binary search time complexity for O (lgN) in the large scale HSIs with a disorder matrix, but more storage space is required. The HD map $p_{ij}$ and the $N_l$-dimensional map $q_{ij}$ are re-defined as follows:

$$p_{ij} = \frac{\exp\left(-\frac{d_x(x_i,\ x_j)^2}{2\sigma^2}\right)}{\sum_{k \in \lambda_i} \exp\left(-\frac{d_x(x_k,\ x_i)^2}{2\sigma^2}\right)}, \qquad q_{ij} = \frac{\exp\left(-d_y(y_i,\ y_j)^2\right)}{\sum_{k \in \lambda_i} \exp\left(-d_y(y_k,\ y_i)^2\right)}. \tag{10}$$

The minimum *KL* divergence of the DR transformation map is determined by a gradient descent algorithm, which is divided into two parts

$$KL_{min} = \frac{\partial C(D_Y)}{\partial y_i} = 4 \sum_{j \neq i} \left(p_{ij} - q_{ij}\right) q_{ij} \tau \left(y_i - y_j\right)$$
$$= 4 \left(\sum_{j \neq i} p_{ij} q_{ij} \tau \left(y_i - y_j\right) - \sum_{j \neq i} q_{ij}{}^2 \tau \left(y_i - y_j\right)\right) = 4 \left(C_{attr} - C_{rep}\right). \tag{11}$$

Herein, $C_{attr}$ and $C_{rep}$ represent the sum of the attractive correlation and repulsive correlation, respectively. *KLmin* evaluates the loss of information between the HD *P* map and LD *Q* map using the SNE iterative process. The lower the *KLmin* value is, the higher the similarity of *P* and *Q* is and the better $D_Y$ represents $D_X$. The $C_{attr}$ subsection computation complexity is O (N) but that of $C_{rep}$ is O ($N^2$), aiming to reduce the $C_{rep}$ computational cost. The Barnes–Hut algorithm [38] is used to approximate $C_{rep}$ with O (NlogN), which is superior to the original algorithm efficiency of the joint distributions *P* and *Q* with O ($N^2$).

Additionally, the interclass correlation coefficient (ICC) algorithm [39] is used for the similarity evaluation between the Gaussian center $x_i$ and the adjacent data points $x_j$ in the interclass of the datasets. We assume that there are *M* sample units $U_i{}^M = \{x_i\}_{i=1}^M \subset \mathcal{R}^{N_l}$ in an interclass of similar points, $x_j \in D_x$, $x_j$ represents the neighboring data points of center $x_i$ in the same sample unit. The $x_i$ and $x_j$ are considered *N* pairwise data points $(x_i, x_j)$, $i \neq j$ and $i, j = 1, \ldots, N$. We assume that $G_i$, $G_d$, and $R_c$ denote the interclass data points, the different class data points, and the ICC, respectively.

$$R_c = \frac{\sum_{i,j=1}^N (x_i - \overline{x})(x_j - \overline{x})}{N\sigma_s^2}, \ R_c \in [0,1] \tag{12}$$

where

$$\overline{x} = \frac{\sum_{i,j=1}^N (x_i + x_j)}{2N}, \ \sigma_s^2 = \frac{\sum_{i=1}^N (x_i - \overline{x})^2 + (x_j - \overline{x})^2}{2N}. \tag{13}$$

If $R_c < 0.5$, $x_i \in G_d$, $x_i$ is considered a singularity and removed from the class. If $R_c \geq 0.75$, $x_i \in G_i$, which can keep in the interclass as nearest similarity points (the threshold of $R_c$ is an empirical value). If $0.5 \leq R_c < 0.75$, there is uncertainty that depends on the required accuracy. In practical applications, the $R_c$ threshold is chosen based on the spectral space density. Generally, the higher the ICC $R_c$ is, the more similar the spectral features of the data points are. The data points are used and classified into the same class. The sketch map of the similarity evaluation using the ICC is shown in Figure 3.

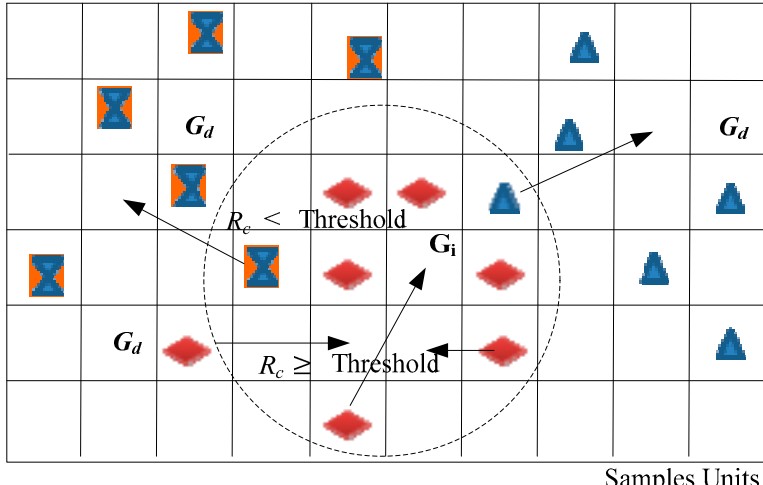

**Figure 3.** Evaluation of similarities of output datasets by ICC.

In the improved t-SNE method, the LR method is embedded to improve the predictability and spectral feature extraction of the DR algorithm due to limited labeled samples. The LR method accelerates the convergence of the DR algorithm and improves the execution efficiency of the spectral feature extraction and classification in the LD data. Compared to the t-SNE, the improved t-SNE model has lower time complexity, faster similarity evaluation, and higher accuracy for interclass matching. The improved method achieves DR, spectral feature extraction, and classification simultaneously. The processing framework of the improved method is shown in Figure 4. The LD data are considered input data for the spatial feature and edge extraction.

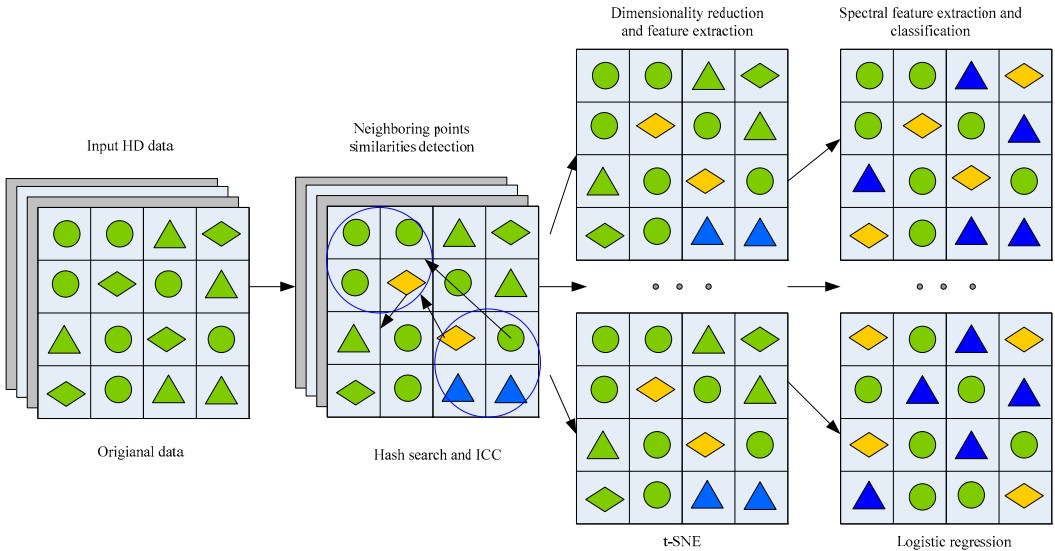

**Figure 4.** The processing diagram of DR and spectral feature extraction by using the improved t-SNE.

## 2.4. CDBNs

The CDBNs method is a hierarchical generative model for deep learning and consists of multiple layers of CRBM. It has been demonstrated that the method has a good performance for remote sensing image interpretation and land object identification. It is well suited for high-level feature learning and scales well to 2-D imagery. The CDBNs method is translation invariant and uses the spatial relationships between neighboring pixels via the CRBM stacked model. A CRBM stacked model is a probabilistic, multi-layer, bipartite graphical model composed of three layers of sets of units, namely an input layer with visible units ($v$), a detection layer with hidden units ($h$), and a pooling layer with

probabilistic max-pooling ($p$). The visible layer and hidden layer are connected symmetrically by the weight matrix $W$. The visible layer consists of a $N_V \times N_V$ ($N_V^2$) matrix of binary units, which represent the results of the algorithm. The detection layer and pooling layer have $K$ groups, each of the groups of hidden layers consists of a $N_H^2$ matrix of binary units and the pooling layer has $N_P^2$ binary units. Each of the $K$ groups has $N_W^2$ feature filter units, which connect the visible layer and the hidden layer. Every filter is regarded as a channel with computing independence, including the convolution and pooling operations. The detection layers compute responses by convolving a feature detector with the previous layers. The pooling layers use a max-pooling method to shrink the representations originating from the detection layers to obtain small translation-invariant input datasets. The layers have a bidirectional feed-forward structure with bottom-up and top-down inference in the generative model. The probability of the two kinds of filter units can be calculated using multiple convolution operations and is defined by the energy equation $E(v, h)$ as follows:

$$\mathcal{P}(v, h) = \frac{exp(-E(v, h))}{\mathcal{Z}}, \tag{14}$$

$$
\begin{aligned}
E(v, h) &= -\sum_{k=0}^{K-1}\sum_{i,j=0}^{N_H-1}\sum_{r,s=0}^{N_w-1} h_{ij}^k W_{rs}^k v_{\lambda,\xi} - \sum_{i=0}^{N_V-1} c_i \sum_{i,j=0}^{N_V-1} v_{ij} - \sum_{k=0}^{K-1} b_k \sum_{i,j=0}^{N_K-1} h_{ij}^k \\
&= -\sum_{k=0}^{k-1} h^k \cdot \left(\widetilde{W}^k * v\right) - \sum_{i=0}^{N_V-1} c_i \sum_{i,j=0}^{N_V-1} v_{ij} - \sum_{k=0}^{K-1} b_k \sum_{i,j=0}^{N_K-1} h_{ij}^k.
\end{aligned} \tag{15}
$$

where $\mathcal{Z}$ is a partition function, $\lambda = i + r - 1$, $\xi = j + s - 1$. The other notations and variables are shown in Table 4.

**Table 4.** Description of the above-mentioned variables and symbols.

| Symbol | Description |
|---|---|
| $N_V^2$ | Number of binary units in visible layer |
| $N_H^2$ | Number of binary units of per group in hidden layer |
| $N_w^2$ | Number of filter weights ($N_w \triangleq N_V - N_H + 1$) |
| $N_P^2$ | Number of shrinking filter units of per group ($N_P = \frac{N_H}{C}$, $C$ is a constant factor) |
| $c_i$ | Bias term of visible units |
| $b_k$ | Bias term of hidden units of per group |
| $W_{rs}^k$ | Weights of symmetric connections of visible layer and hidden layer |
| $\widetilde{W}^k$ | Horizontally and vertically flipped array of $W^k$ |
| $\cdot$ | Element-wise product followed by summation |
| $*$ | Convolution |
| $p_\alpha^k$ | A binary unit in the pooling layer, connecting a block $\alpha$ in the detection layer |
| $\beta_\alpha$ | Range of pooling $\beta_\alpha \triangleq \left\{(i, j), h_{ij} \text{ belongs to the block } \alpha.\right\}$ |

The block Gibbs sampling is represented by a conditional distribution of the convolution operation as follows:

$$
\begin{aligned}
\mathcal{P}\left(h_{ij}^k = 1 \middle| v\right) &= sigmoid(\left(\widetilde{W}^k * v\right)_{ij} + b_k), \\
\mathcal{P}\left(v_{ij} = 1 \middle| h\right) &= sigmoid\left(\left(\sum_{k=0}^{k-1} W^k * h^k\right)_{ij} + c_i\right).
\end{aligned} \tag{16}
$$

The conditional probability is re-defined using the max-pooling method

$$
\begin{aligned}
\mathcal{P}\left(h_{i,j}^k = 1 \middle| v\right) &= \frac{exp\left(\mathcal{E}\left(h_{i,j}^k\right)\right)}{1 + \sum_{(i',j') \in \beta_\alpha} exp\left(\mathcal{E}\left(h_{i',j'}^k\right)\right)}, \\
\mathcal{P}\left(p_\alpha^k = 0 \middle| v\right) &= \frac{1}{1 + \sum_{(i',j') \in \beta_\alpha} exp\left(\mathcal{E}\left(h_{i',j'}^k\right)\right)}, \quad \mathcal{E}\left(h_{i,j}^k\right) \triangleq b_k + (\widetilde{W}^k * v)_{ij}.
\end{aligned} \tag{17}
$$

In CBDNs, each layer is greedily trained from lowest to highest [29] and while a layer is being trained, the weights of the interlayer connection are fixed. Subsequently, this layer becomes the input

for the next layer. A model with multi-layer max-pooling can effectively learn higher-level features using convolution and pooling operations. This results in a hierarchical generative model and allows for edge detection in the images using sparse regularization [40]. Therefore, CBDNs method is capable of efficiently extracting features from images, including spatial features and edge information.

### 2.5. d-CDBNs

The CDBNs method provides scaling of full-sized images using PMP by shrinking the representation of the higher layers. It improves the efficiency by weight sharing among all locations of images. The weight sharing provides a symmetrical connection between the visible layer and the detection layer. The shared weights are updated once a minimum difference between the reconstructed data and the input data has been reached. The weight matrix influences the classification results of the CRBM's three layers and there exists a certain redundancy that controls the value of the energy function and the probability distribution. Therefore, it is important for further optimize the weights.

We propose a weight shrinking strategy to reduce the weight redundancy of the interlayers using a deep compression method [41]. The weight-shrinking strategy has three steps: (1) network pruning [42] removes the connections with weights that fall below a certain threshold and a sparse matrix index is created. (2) Weight quantization and weight sharing further compress the pruned network and decrease the number of weights by weight sharing of multiple connections using k-means clustering, thereby fine-tuning the shared weights. (3) Huffman coding is used to compress the quantization value without loss of data. The deep compression weight strategy is embedded in the CDBNs to create the d-CDBNs method.

In the d-CDBNs, the weight quantization and sharing are achieved using the k-means cluster method. This step ensures that all the weights are in the same cluster and the same weights occur in the trained network. The original sub-weights are defined as $W^s = \{w_1^s, w_2^s, \ldots, w_m^s\}$. These are divided into $\kappa$ cluster centers with linear initialization $C = \{c_1, c_2, \ldots, c_\kappa\}$ ($\kappa \ll m$). The minimum sum of squares in a cluster is defined as follows:

$$\underset{C}{argmin} \sum_{i=1}^{\kappa} \sum_{w^s \in c_i} |w^s - c_i|^2. \tag{18}$$

The $\kappa$ clusters require $\theta = \log_2(\kappa)$ bits for index encoding and $\kappa$ shared connection weights in a network with $n$ layer connections, where each connection represents $t$ bits. The compression ratio $\mathcal{R}_{cpr}$ of the quantization is given by:

$$\mathcal{R}_{cpr} = \frac{nt}{n\theta + \kappa t}. \tag{19}$$

The compression ratio $\mathcal{R}_{cpr}$ reflects the efficiency of the connection of the shared weights and represents the original connection during the quantization process. Quantization and training ensure that the number of weights is reduced. To achieve further compression, Huffman coding can provide a reduction of 20–30% in the storage requirements of the connection weights [41]. The deep compressed weights are considered the shared weights of the connection layer. The d-CDBNs method inherits all the functions of the CDBNs method plus the compressed weight sharing. The processing framework is illustrated in Figure 5. The input data are the results of the improved t-SNE model. In the CRBM layer, the number of shared weights is reduced by deep compression, which improves the computational efficiency and reduces space complexity. In addition, the d-CDBNs method can extract deep spatial features and learn edge detection, and these results are then combined with the spectral features to improve HSIs classification accuracy.

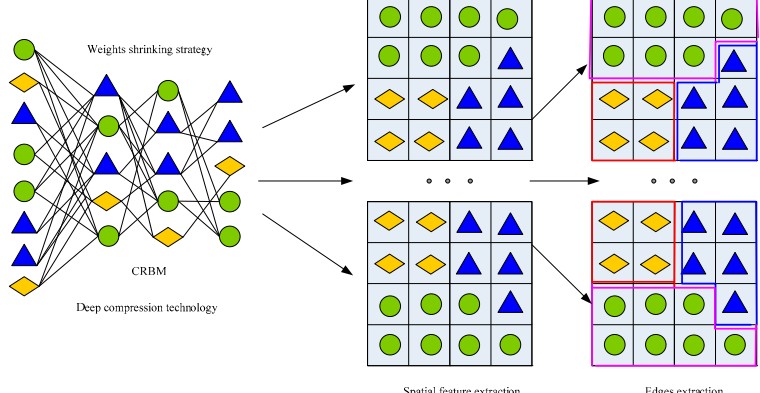

**Figure 5.** The processing framework of weights shrinking, spatial feature and edges extraction by d-CDBNs.

## 2.6. MFEM

The MFEM model is a combination of the nonlinear DR technique, the improved t-SNE, the d-CDBNs method, and the LR classifier. It represents a novel UIS detection model for HSIs and provides multi-feature extraction. The multi-feature allows for the effective identification and extraction of impervious surfaces in each pixel. These features include spectral signatures, spatial features, and edge information. The spectral signatures are the reflectance values of each pixel in every spectral band; the spatial features reflect the structure, shape, and texture of image objects; and the edge information describes differences between local attributes and adjacent attributes at the boundary of different landscapes. The edge information includes the specific properties of an object that differentiates it from other objects. The MFEM has three major processing steps, as shown in Figure 6: (1) DR and spectral feature extraction using the improved t-SNE model; (2) spatial feature extraction and edge information learning using the d-CDBNs; and (3) multi-feature extraction and LR classification. Compared with the standard t-SNE and the CDBNs methods, the MFEM model is characterized by a faster neighborhood search, higher accuracy for interclass point matching, less shared weight redundancy, and better predictability. Additionally, the multi-feature improves the image interpretation and UIS detection.

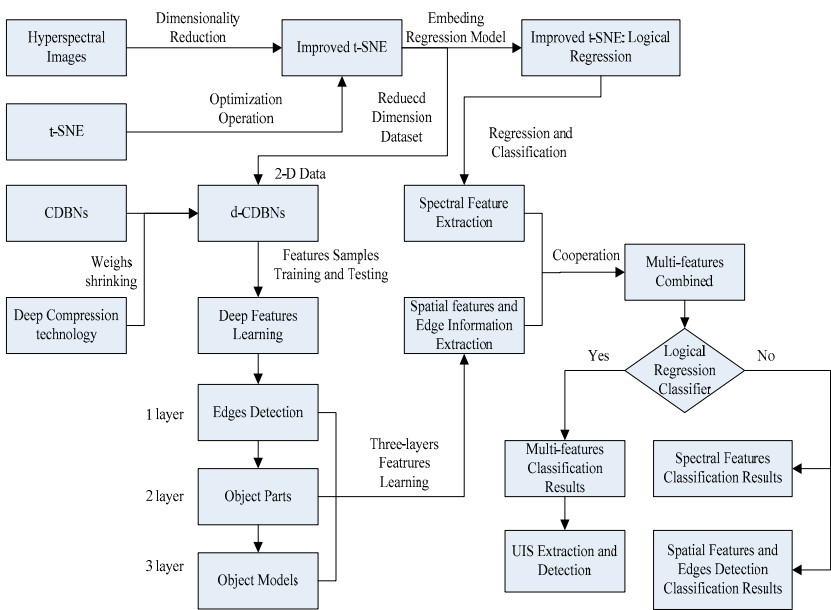

**Figure 6.** Framework of the MFEM model.

## 3. Results

For the performance evaluation of the MFEM method, we used common evaluation indices including the producer's accuracy (PA), user's accuracy (UA), kappa coefficient (Kappa), overall accuracy (OA), and average accuracy (AA) (see Abbreviations). All experiments were performed using MATLAB R2016a on a Windows 7 (64-bit) Intel Core i5-4200 3.2 GHz 8 GB RAM machine.

### 3.1. Evaluation of the Improved t-SNE Method

### 3.1.1. Perplexity Configuration

In the improved t-SNE method, the perplexity ($P_i$) increases with increases in the Shannon entropy and increases the Gaussian variance $\sigma_i$. This results in an increase in the number of nearest neighbor points. To determine the optimal $\sigma_i$ and obtain a suitable number of nearest neighbor points, we used the hash table search to replace the binary search algorithm. In the experiment, the perplexity ($P_i$) values were set at 10, 20, 30, 40, and 50, respectively, due to the larger number of discrete points, the unstable OA, and the poor robustness when $P_i < 5$ or $P_i > 50$. The dimensional space of the output embedded data was set as the default 2-D, the number of iterations was 1000, the correlation parameter was 0.75, and the minimum gradient descent was $1 \times 10^{-7}$. The ground-truth datasets were considered the reference for creating the confusion matrix. The evaluation indicators were used to assess the performance of the improved t-SNE method for the two datasets (Table 5). The two datasets had different results when different $\sigma_i$ were used for the perplexities. The results show that the values of the evaluation indices increase with the increase in the perplexity $P_i$. However, the mean $\sigma_i$ exhibits a decreasing rate of growth once the perplexity's predefined threshold value is reached. This also demonstrates that the optimal value range of the perplexity is from 5 to 50. The classification accuracy is higher for the city center scene than the university scene. The reason is that the city center scene has more reference data points than the university scene.

**Table 5.** Performance evaluation of improved t-SNE with different perplexity for spectral features extraction.

| Perplexity ($P_i$) | University Scene | | | | City Center Scene | | | |
|---|---|---|---|---|---|---|---|---|
| | Mean $\sigma_i$ | OA (%) | AA (%) | Kappa | Mean $\sigma_i$ | OA (%) | AA (%) | Kappa |
| 10 | 1.506 | 77.67 | 75.94 | 0.6367 | 1.159 | 80.34 | 78.52 | 0.6851 |
| 20 | 1.663 | 78.56 | 77.29 | 0.6523 | 1.663 | 81.58 | 79.87 | 0.6979 |
| 30 | 1.751 | 80.98 | 78.48 | 0.6761 | 1.917 | 82.21 | 80.55 | 0.7087 |
| 40 | 1.815 | 82.78 | 79.77 | 0.6832 | 2.114 | 83.67 | 81.63 | 0.7256 |
| 50 | 1.865 | 83.72 | 80.35 | 0.6897 | 2.277 | 85.62 | 82.79 | 0.7472 |

In addition, the number of dimensions (after executing the DR) influences the classification accuracy. We tested dimensions from 2 to 12 and perplexity values from 10 to 100 for the two datasets, as shown in Figure 7. The OA of the two datasets exhibits different degrees of increase with an increasing number of dimensions. The results are similar for the different perplexity values.

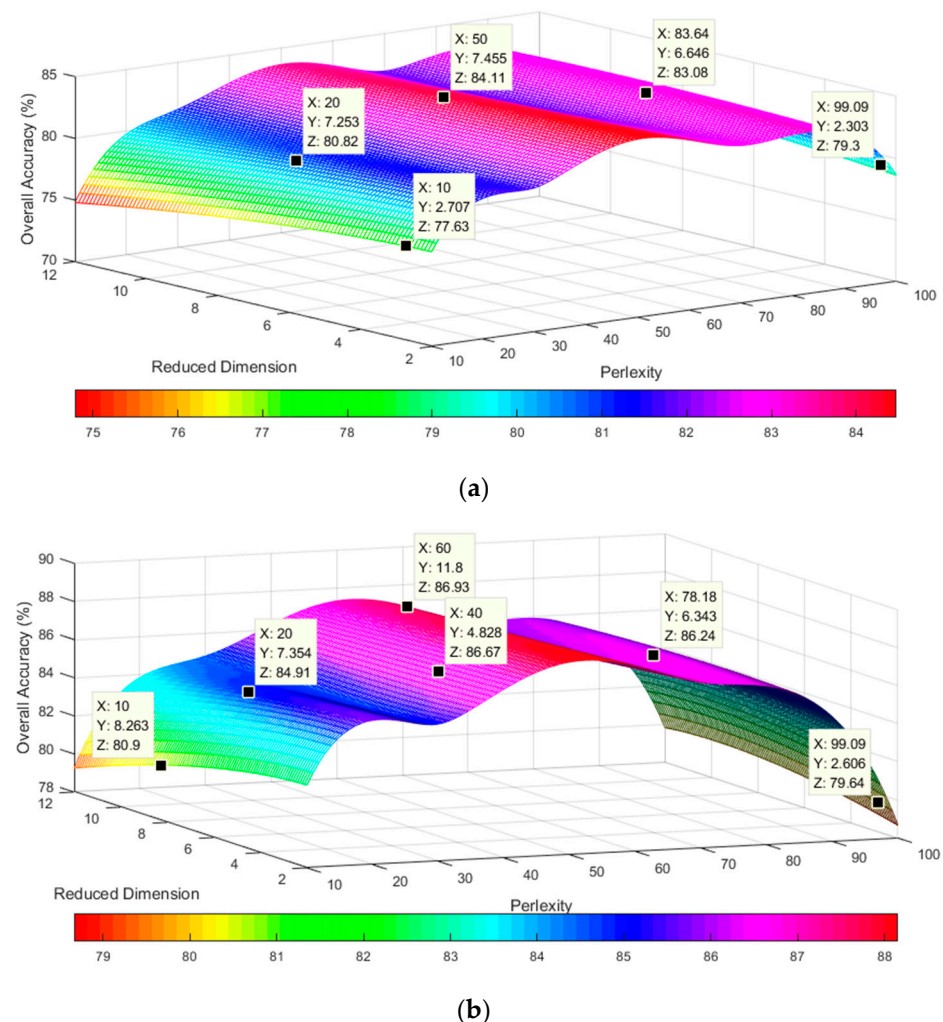

**Figure 7.** Performance evaluation of the improved t-SNE method for two datasets with different dimensions and perplexity values (x-axis, reduced dimension; y-axis, perplexity; z-axis, overall accuracy): (**a**) university scene; and (**b**) city center scene.

### 3.1.2. Computation Time

In the DR process, the major time cost is the search for the nearest neighbor points and the LD map embedding. To optimize the algorithm time complexity, the hash table search algorithm was implemented in the improved t-SNE and replaced the binary algorithm. We tested various perplexities and dimensions using the t-SNE method and improved t-SNE method for the two datasets, as shown in Table 6 and Figure 8. For the university scene, the average computation time is 10.85% less for the improved t-SNE method than the standard t-SNE method. However, the average computation time is only 4.11% less for the city center scene because of the larger data volume of the dense buildings. Overall, the computation time is lower for the improved t-SNE then the standard t-SNE, although the computation time increases for both methods and both datasets with an increase in the number of perplexities. This reason for this result is that the complexity is lower for the hash table search method than the binary search method. In addition, the minimum *KL* divergence process reduces the time complexity due to the Barnes–Hut algorithm. An increase in the perplexity increases the number of nearest neighbor points of the kernel points. Additionally, the computational time cost is higher for the city center scene than the university scene due to the larger building density of the city center scene.

**Table 6.** Computation time of improved t-SNE and t-SNE for university scene (U) and city center scene (C).

| Perplexity ($P_i$) | Dimension | University Scene | | City Center Scene | |
|---|---|---|---|---|---|
| | | t-SNE(U) | Improved t-SNE(U) | t-SNE(C) | Improved t-SNE(C) |
| 10 | 2 | 133.56 | 121.12 | 242.28 | 231.19 |
| 30 | 4 | 153.72 | 133.45 | 247.36 | 233.77 |
| 50 | 6 | 162.58 | 141.72 | 248.71 | 242.38 |
| 70 | 8 | 167.03 | 149.88 | 256.39 | 245.46 |
| 90 | 10 | 175.78 | 157.61 | 265.59 | 253.36 |
| 100 | 12 | 184.64 | 167.52 | 271.62 | 262.85 |

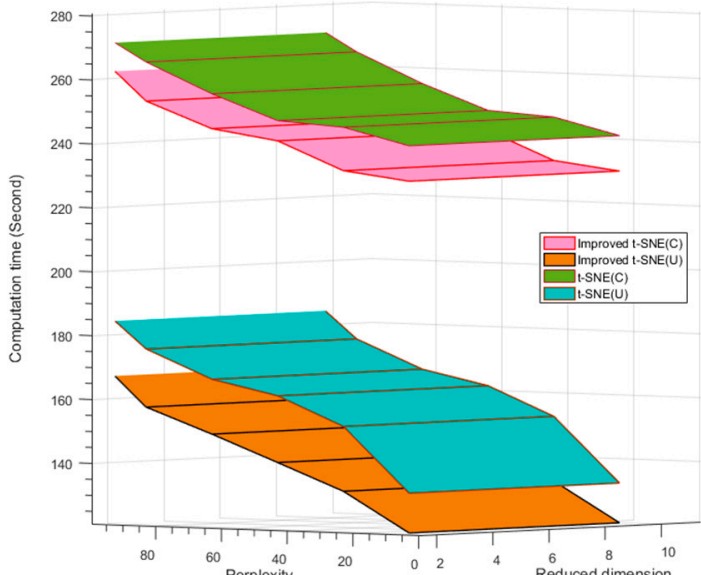

**Figure 8.** Comparison of computation time between improved t-SNE and t-SNE for university scene (U) and city center scene (C).

### 3.1.3. Interclass Correlation Configuration

The ICC algorithm evaluates the similarities of the nearest neighbor points and reduces the classification error in the interclass. For the evaluation of the ICC performance of the improved t-SNE model, we set the perplexity as 35 and the DR of the result dataset to 2. The reduced dimension dataset has the embedded LR algorithm and the interclass correlation $R_c$ has a range of 0.2–0.95. The accuracy of the nearest neighbor points influences the spectral extraction and classification results (Figure 9). As shown in Figure 9a, the OA of the two datasets increases sharply when $R_c < 0.6$ and the results are better for the city center scene than the university scene when $R_c > 0.6$. The highest OA of the city center scene is 93.57%, which is 5.82% higher than the OA of the university scene. As shown in Figure 9b, the classification error rate decreases with an increase in the value of the ICC $R_c$. The lowest classification error is 0.106 when $R_c = 0.95$ for the city center scene, i.e., the OA of similarities of nearest neighbor points achieves 89.4%. To minimize the influence of the threshold, we used $R_c = 0.85$, which provides optimal efficiency in the experiments (see Section 4 for the reasons).

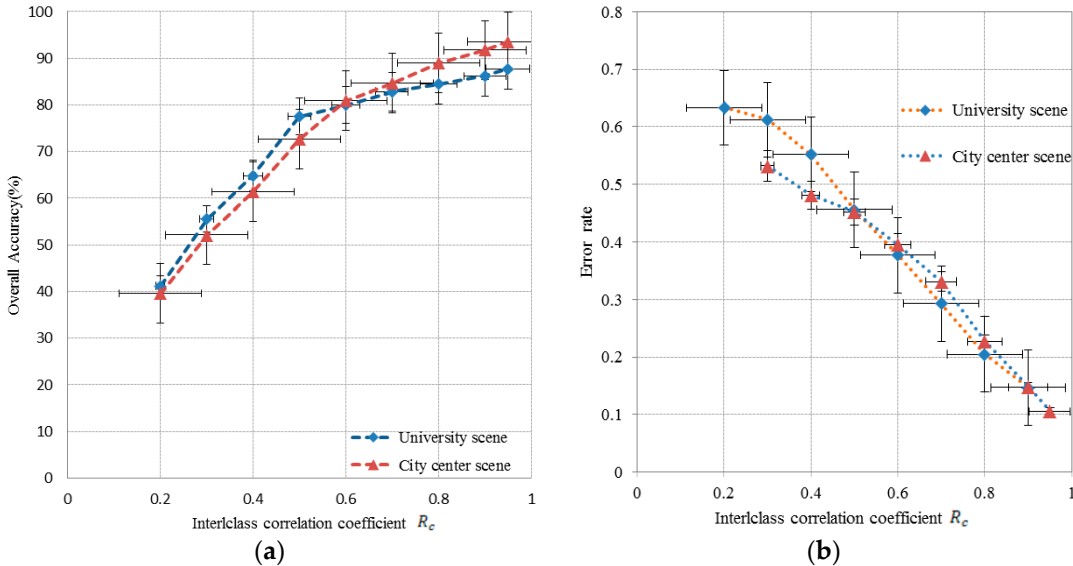

**Figure 9.** Different results of classification and neighbor points error with interclass correlation coefficient $R_c$: (**a**) overall accuracy with change of $R_c$; and (**b**) interclass classification error rate with change of $R_c$.

### 3.2. Comparison of the Spectral Feature Extraction Results with Other DR Methods

The improved t-SNE model has an embedded LR algorithm to improve the classification performance and predictability of the samples. This process allows for sequential spectral feature extraction for LD datasets and improves the classification accuracy. The results of the improved t-SNE model were compared with those of other frequently-used DR algorithms (Table 7), including the supervised algorithms LDA, LDE, and LFDA [43] and the unsupervised algorithms PCA and the standard t-SNE. The dimension was 2-D and the number of perplexities was 30. For the unsupervised algorithms, we selected 20 training samples randomly for each dataset and 260 unlabeled samples as test samples for each class. Table 7 shows that the improved t-SNE model outperforms the other DR algorithms in terms of classification accuracy. All supervised DR algorithms have higher OA than the PCA. Specifically, the LFDA is more effective than the other supervised DR algorithm for the hyperspectral datasets. Under the same conditions, the improved t-SNE achieves a higher OA than the standard t-SNE and the accuracy improvements are 4.22% and 4.08%, respectively, for the university scene and city center scene.

**Table 7.** Comparison of results assessment: both improved t-SNE method and other DR methods for spectral features extraction.

| Dataset | Evaluation Index | LDA | LDE | LFDA | PCA | t-SNE | Improved t-SNE |
|---|---|---|---|---|---|---|---|
| University scene | PA (%) | 73.71 | 73.93 | 76.74 | 72.78 | 76.69 | 80.38 |
| | UA (%) | 72.84 | 72.75 | 75.81 | 72.35 | 76.55 | 79.89 |
| | OA (%) | 73.49 | 73.85 | 76.17 | 72.68 | 76.41 | 80.63 |
| | AA (%) | 72.75 | 72.56 | 75.44 | 71.57 | 75.64 | 79.86 |
| | Kappa | 0.7183 | 0.7171 | 0.7532 | 0.7014 | 0.7445 | 0.7771 |
| City center scene | PA (%) | 68.43 | 70.85 | 75.32 | 64.71 | 77.89 | 82.36 |
| | UA (%) | 66.87 | 70.26 | 74.33 | 63.27 | 77.05 | 81.76 |
| | OA (%) | 68.72 | 71.22 | 75.13 | 64.56 | 78.55 | 82.63 |
| | AA (%) | 65.88 | 70.76 | 74.26 | 64.13 | 77.83 | 82.04 |
| | Kappa | 0.6796 | 0.7103 | 0.7515 | 0.6422 | 0.7713 | 0.8183 |

The OA results for the different methods and reduced dimensionality values of 2–12 are shown in Figure 10. It is evident that the OA increases for all methods as the dimensionality decreases, i.e.,

the values increase from 2 to 12. In the university scene dataset, the mean OA of the improved t-SNE is 85.07%, which is 3.827% higher than the mean value of the other DR methods. In the more complex city center scene dataset, the mean OA of the improved t-SNE is 8.747% higher than that of the other DR methods. The reason is the lower performance of the other methods in the more complex city center scene, whereas the improved t-SNE is less affected in accuracy by the complexity.

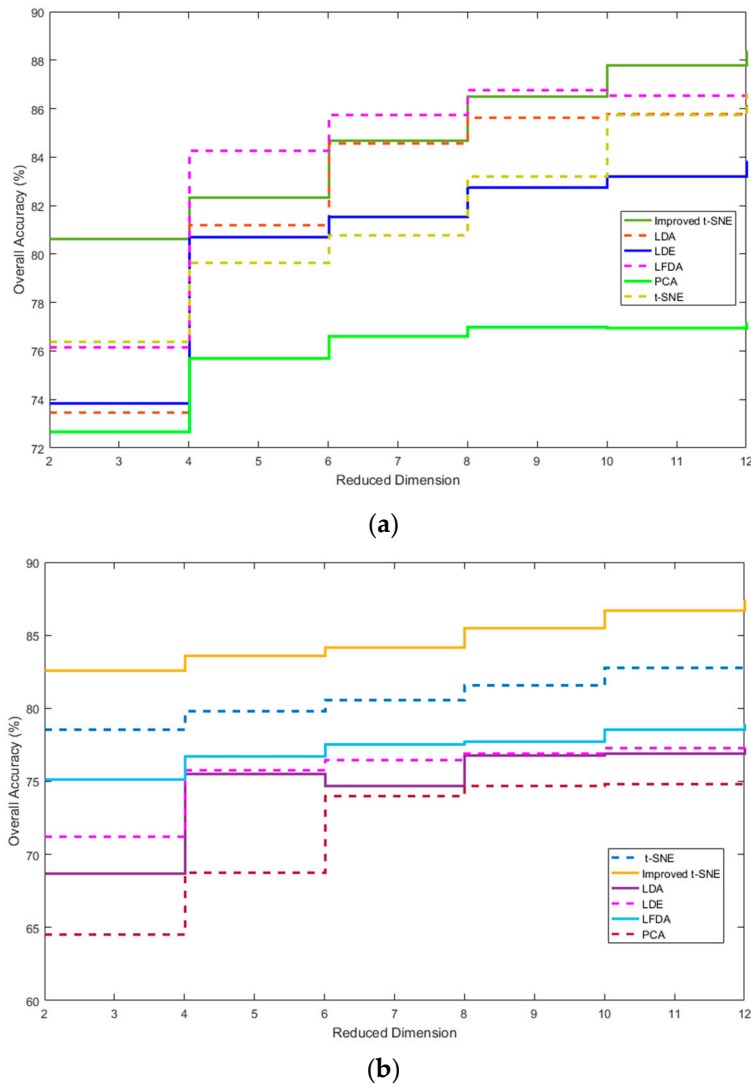

**Figure 10.** Comparison of different DR algorithms with various reduced dimensions, from 2 to 12: (**a**) university scene; and (**b**) city center scene.

However, the classification results of the spectral extraction retain the "salt and pepper" noise effect and misclassification. The main reason of misclassification for areas with similar reflectivity values, such as confusion between shadow and asphalt in the university scene and confusion between bare soil and brick in the city center scene. This appearance needs to be counteracted by the spatial feature extraction.

### 3.3. Verification of the d-CDBNs Method

In the aforementioned experiment, we examined the performance of the improved t-SNE model and achieved good results. After using the improved t-SNE DR model, the two datasets were translated into a 2-D representative matrix. The results of the DR processing and spectral extraction were inputs for the d-CDBNs method. The d-CDBNs method consists of learning and training using three layers.

In the first layer, i.e., the input layer, the university scene consists of 4148 binary units and the city center scene consists of 8160 binary units. In the second layer, the hidden layer consists of $k = 32$ groups and each group has $N_H^2 = 130$ and $N_H^2 = 255$ hidden units for the university scene and city center scene, respectively. In the third layer, i.e., the pooling layer, we used a shrink factor of $C = 2$ and there were 65 and 128 binary units for the two scenes, respectively.

In the MFEM, the d-CDBNs method was used to extract the spatial features and edges. These features are then combined with the spectral features for a multi-feature UIS classification method. The multi-feature combination reduces the "salt and pepper" effect. To examine the performance of the d-CDBNs method, we randomly selected about 15% of the reference image pixels as training data and used the remaining 85% as testing data. We set the pixel filters for the $5 \times 5$ matrices and the learning rate was 0.05. The number of connection weights between the visible layer and the detection layer was set at $N_w^2 = 40$ and $N_w^2 = 66$ for the university scene and city center scene datasets, respectively, in the first layer. The other parameters were the default values.

### 3.3.1. Compression of the Shared Weights of the Layers for Spatial Feature Extraction

For testing the performance of the shared weight deep compression in the d-CDBNs method, we compared the results with that of the standard CDBNs method in terms of network connection number, number of weights, data size, and computation time (Tables 8 and 9). In the d-CDBNs method, we set the compression ratio $\mathcal{R}_{cpr}$ at 4× and used two hidden layers for detecting and convolution. The results indicated that the d-CDBNs effectively reduce the number of network connections, shared weights, the data size, and the computation time. We only embedded the deep compression method in the first layer due to time and complexity considerations. Therefore, the number of weights is the same in the first layer and the second layer. In the third layer, the max-pooling shrinks the representation of the detection layer, which reduces the computational cost. The reason for the lower number of weights in the second layer of the CDBNs is that the shared weights replace part of the initial weights. After weight compression, the computation time and memory requirements are significantly lower, and do not negatively affect the accuracy of results, which was verified by Han et al. [42].

**Table 8.** Comparison of performance between CDBNs and d-CDBNs for university scene.

| Layer | CDBNs | | | | d-CDBNs ($\mathcal{R}_{cpr}$ = 4) | | | |
|---|---|---|---|---|---|---|---|---|
| | No. of Connections | No. of Weights | Data Size | Computation Time (Second) | No. of Connections | No. of Weights | Data Size | Computation Time (Second) |
| 1 | 130 | 40 | 25 M | 186.4 | 114 | 10 | 6.25 M | 98.3 |
| 2-1 | 130 | 32 | 25 M | 136.8 | 114 | 8 | 6.25 M | 72.8 |
| 2-2 | 130 | 24 | 25 M | 125.7 | 114 | 5 | 6.25 M | 55.5 |
| 3 | 65 | 15 | 15.6 M | 95.6 | 57 | 3 | 3.56 M | 36.9 |

**Table 9.** Comparison of performance between CDBNs and d-CDBNs for city center scene.

| Layer | CDBNs | | | | d-CDBNs ($\mathcal{R}_{cpr}$ = 4) | | | |
|---|---|---|---|---|---|---|---|---|
| | No. of Connections | No. of Weights | Data Size | Computation Time (Second) | No. of Connections | No. of Weights | Data Size | Computation Time (Second) |
| 1 | 255 | 66 | 64 M | 202.9 | 223 | 17 | 16 M | 115.5 |
| 2-1 | 255 | 47 | 64 M | 179.3 | 223 | 14 | 16 M | 94.6 |
| 2-2 | 225 | 26 | 64 M | 135.6 | 223 | 10 | 16 M | 73.3 |
| 3 | 128 | 18 | 35.8 M | 110.5 | 112 | 5 | 8.4 M | 45.6 |

### 3.3.2. Edge Detection

In the d-CDBNs, the first layer detects the edge information of the input dataset and the edges of the image objects are extracted in the second layer. We used a $5 \times 5$ pixel filter for the edge detection followed by the sparse regularization method, which was proposed by Lee et al. [40]. As shown in Figure 11, the results of detecting the edges in the university dataset are excellent and corners, contours, edge angles, and boundaries are extracted.

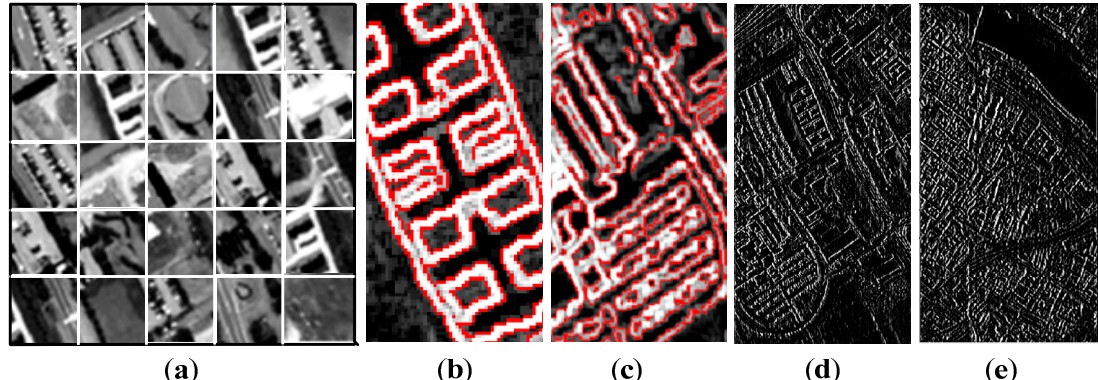

**Figure 11.** Edges detection via sparse regularization: (**a**) 5 × 5 pixels filter units; (**b**) sparse land cover edges; (**c**) dense land cover edges; (**d**) university scene; and (**e**) city center scene.

*3.4. Accuracy Comparison of MFEM Model for Impervious Surface Detection*

The MFEM model has three major functions: DR and spectral feature extraction, spatial feature and edge extraction, and multi-feature classification. To evaluate the MFEM model performance for UIS extraction, we compared the results with that of popular machine learning methods, including an SVM, CNNs, DBNs, and CDBNs. The input data for these methods are the results of the improved t-SNE. The classification accuracy results are shown in Table 10. It is evident that the MFEM model has higher accuracy than the other machine learning methods. The MFEM model enhances the OA by 6.78% (university scene) and 5.19% (city center scene), comparing with average of other methods, as well as the Kappa by 6.81% (university scene) and 6.21% (city center scene). The classification accuracy is higher for the university scene than the city center scene. The reason is that the university scene has more concise outlines and clearer boundaries than the city center scene, which results in higher accuracy for the spatial feature extraction and edge detection.

**Table 10.** Evaluation of classification accuracy of MFEM model compares with other commonly used algorithms.

| Datasets | Evaluation Index | SVM | CNNs | DBNs | CDBNs | MFEM |
|---|---|---|---|---|---|---|
| University scene | OA (%) | 89.35 | 93.48 | 91.23 | 94.27 | 98.87 |
| | AA (%) | 88.65 | 93.11 | 90.77 | 94.89 | 98.35 |
| | Kappa | 0.8621 | 0.9171 | 0.9017 | 0.9325 | 0.9714 |
| City center scene | OA (%) | 88.62 | 92.22 | 90.95 | 93.89 | 96.61 |
| | AA (%) | 88.09 | 91.92 | 90.25 | 93.16 | 96.84 |
| | Kappa | 0.8617 | 0.9045 | 0.8813 | 0.9149 | 0.9527 |

The results of the UIS extraction were compared with those of the above-mentioned methods (Figure 12). It was observed that, in the commonly used methods, the lower classification accuracy stems from the confusion between classes, such as bare soil and brick in the university scene and sediments in the shoal water and asphalt in the city center scene. The multi-feature extraction in the MFEM model reduces the confusion and minimizes the "salt and pepper" effect. In Figure 12, from left to right are SVM, CNNs, DBNs, CDBNs and MFEM (see Abbreviations) results, respectively. At the bottom of each comparison figure, we added part of the zoom image to clarify the algorithm performance.

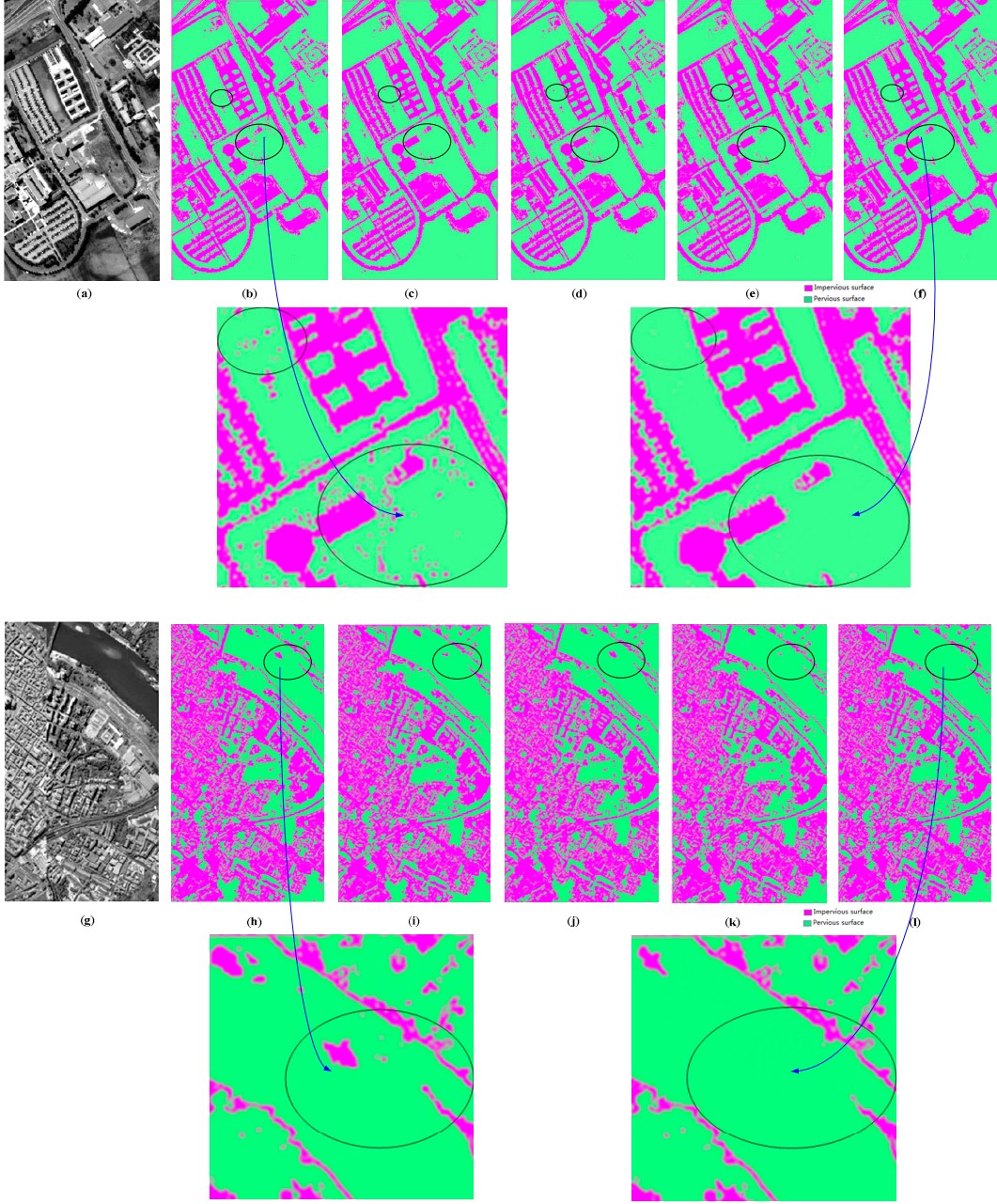

**Figure 12.** Results of impervious surface extraction use by successively SVM, CNNs, DBNs, CDBNs, and MFEM: (**a**, **g**) single band original image; (**b**–**f**) results of university scene; and (**h**–**l**) results of city center scene.

## 4. Discussion

In this paper, we propose the novel MFEM model based on multi-feature extraction for UIS detection from hyperspectral datasets. The model combines a nonlinear DR with the improved t-SNE and the deep learning d-CDBNs method. The improved t-SNE method is used to translate the HD data into LD data and extracts the spectral features. In the experiment, we used different dimension number and perplexity to test the two datasets, respectively, and explored the influence on the overall accuracy. We found that the overall accuracy would be improved with the increase of dimension number, but the influence of the change of perplexity on the overall accuracy would fluctuate. We took the perplexity and reduced dimension as the independent variables, overall accuracy as the dependent variable, built the 3-D surface figure such as Figure 7. The maximum overall accuracy of the university

scene and the city center scene datasets was 84.23% and 87.87%, respectively. The ICC algorithm evaluates the similarities of the nearest neighbor points, improves classification accuracy and reduces the classification error in the interclass. However, a higher interclass correlation threshold has a negative effect on the number of nearest neighbor points and the classification results. First, a higher interclass correlation threshold results in a smaller number of nearest neighbor points, thereby reducing the perplexity. Second, a higher interclass correlation threshold results in fewer nearest points in the interclass of a kernel point. Therefore, if a larger number of Gaussian kernel points are selected, the memory requirements and time cost of the improved t-SNE will increase.

The d-CDBNs method performs spatial feature and edge extraction. The method embedded deep compression strategy for shared weights. The experiments showed that the compression strategy effectively reduced the complexity of data, especially the workload of convolution operation, and improved the performance of d-CDBNs algorithm. The setting of the compression ratio $\mathcal{R}_{cpr}$ affects the compression effect and the accuracy of the algorithm, while the setting of a larger compression ratio will negatively affect the extraction accuracy. Generally, setting the compression ratio at $4\times$ will not negatively affect the accuracy of the algorithm [42].

It is worth mentioning that the shadows in the two datasets were considered pervious surfaces in this experiment. Most shadows occur in meadows or trees in the university scene and are cast by the sparse building. However, in the city center scene, some shadows occur in streets with asphalt, especially near dense and low buildings. These shadows are classified into the pervious surface classes, which caused a moderate reduction in the classification accuracy. The reason can be attributed to two factors: first, narrow streets have vegetation on both sides; and, second, the shadows cast by trees mostly occur in meadows or bare soils in green areas. To achieve a unified classification strategy for both scenes, we considered the shadows in both scenes as pervious surfaces.

## 5. Conclusions

The MFEM model optimizes and improves the standard t-SNE and CDBNs and provides the following improvements: (1) Faster neighborhood point searching and higher accuracy for the interclass point similarity detection for the DR and spectral feature extraction. On average, the MFEM model reduces time by 10.85% (university scene) and 4.11% (city center scene), and enhances the OA by 3.83% (university scene) and 8.75% (city center scene). (2) The LR algorithm increases the predictability for reducing dependence on labeled data. (3) The deep compression algorithm reduces the weight redundancy of the connected convolution layers and improves the computation time and storage requirements. The MFEM model reduces time cost by 88.1 s and 87.4 s, and storage by 18.75 MB and 47 MB for university and city center scene, respectively. (4) The multi-feature extraction combines spectral, spatial and edge extraction of UIS using the LR classifier. The MFEM model achieves the OA of 98.87% (university scene) and 96.16% (city center scene). Therefore, the provided MFEM model has better performance than other commonly used methods in term of efficiency and accuracy of the UIS feature extraction and detection.

**Author Contributions:** Y.W. proposed and designed the MFEM model and wrote the manuscript. Y.W. and H.S. conceived and performed the experiments and analyzed the experimental results. M.L. designed and revised the manuscript.

**Funding:** This work was financially supported by National Natural Science Foundation of China under Grant 31670552, and was performed while the corresponding author acted as an awardee of the 2017 Qinglan Project sponsored by Jiangsu Province.

**Acknowledgments:** The authors would like to thank L. Maaten for sharing the t-SNE source code, Scholar H. Lee for providing the CDBNs source code and toolbox, and the Computational Intelligence Group for providing datasets.

**Conflicts of Interest:** The authors declare no conflict of interest.

## Abbreviations

List of nomenclature in this paper.

| Abbreviation | Description |
| --- | --- |
| AA | average accuracy |
| CDBNs | convolutional deep belief networks |
| CNNs | convolutional neural networks |
| CRBM | convolutional restricted Boltzmann machine |
| DAE | deep auto-encoders |
| DBMs | deep Boltzmann machines |
| DBNs | deep belief networks |
| d-CDBNs | deep compression convolutional deep belief networks |
| DR | dimensionality reduction |
| HD | high-dimensional |
| HSIs | hyperspectal images |
| ICC | interclass correlation coefficient |
| Kappa | kappa coefficient |
| KL | Kullback–Leibler |
| LD | low-dimensional |
| LDA | linear discriminant analysis |
| LDE | local discriminant embedding |
| LFDA | local Fisher discriminant analysis |
| LR | logistic regression |
| MFEM | multi-feature extraction model |
| OA | overall accuracy |
| PA | producer's accuracy |
| PCA | principal component analysis |
| PMP | probabilistic max-pooling |
| SDA | stacked denoising auto-encoders |
| SVM | support vector machine |
| t-SNE | t-distributed stochastic neighbor embedding |
| UA | user's accuracy |
| UIS | urban impervious surfaces |

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
