# Peer review of "An Improved Model Based Detection of Urban Impervious Surfaces Using Multiple Features Extracted from ROSIS-3 Hyperspectral Images"

_remotesensing, doi:10.3390/rs11020136_

Round 1

Reviewer 1 Report

This paper presented a novel MFEM for urban impervious surface extraction. The proposed method clearly brings a novel solution to the problem and nicely written. The results show future applicability of the method, however with scope for future developments. I find the overall solution of MFEM a little to complex, yet very effective. I hope with further research in this direction would make it more simplified.

Author Response

We are extremely grateful for the reviewers’ constructive suggestions. Based on these comments, we presented responding point-by-point replies in the uploaded attachment and have made careful modifications to the original manuscript. All modifications according to reviewers’ comments are marked in blue ink in the revised version

Reviewer 2 Report

Authors want to transfer research done for detection of Urban Impervious Surfaces Using Multiple Features Extracted from Hyperspectral Images and improve the object detection and extraction performances for Hyperspectral Images. The method has multiple applications and would be of interest to different professionals. Given my different background in satellite images processing, I'm not aware of this research is fully innovative and if the literature analysis is satisfactory. My review will be perhaps more focusing on general aspects.

The topic is for sure significant, and the general method seems to be working, confirmed by the validation process. Unfortunately, in my opinion, the current manuscript requires prior revisions in order to make the presented work publishable.

 I found the whole methodology section to be presumptive, in that it was written as if the reader knows all of the individual lingo associated with each step. In general, you want to take your reader through the steps as clearly as possible and don’t assume that the reader knows exactly what terms mean. If you want people to read and use the algorithms you have to make it accessible to them.

I would have preferred to find a "Discussion" paragraph where to condensate many sentences found here and there and including all elements that might give a feeling on how much the technique is general. Without a conscious discussion, results sound very much contingent to the test case and the comparison between the different techniques do not add added value.

I do believe this work deserves to be published but it should first undergo major revisions as specified above before it will be suitable for publication.

Author Response

(The authors gave the same response as above.)

Reviewer 3 Report

Integrate with more details table 1

Author Response

Thank you very much for having our paper entitled " An Improved Model Based Detection of Urban Impervious Surfaces Using Multiple Features Extracted from ROSIS-3 Hyperspectral Images" reviewed and sending us a bunch of comments, which are quite helpful for us to improve the manuscript. We have carefully revised the manuscript in accordance with the comments raised in the peer-review process, and the original comments are presented in black and our corresponding point-by-point replies are presented in red. Additionally, we have checked the entire sections of the manuscript including main text, figures, tables and references to ensure its compliance with the style or format of Remote Sensing. All modifications according to reviewers’ comments are marked in blue ink in the revised version. The itemized response to each comment is provided as follows. 

Point 1:  Integrate with more details table 1

Response 1:

Thanks for the reviewer’s comment. We have carefully checked the proposed model characteristics and compared commonly used methods, and added another two columns in Table 1, to convey more comparable details, as shown in page 3-4.

Round 2

Reviewer 2 Report

I think the authors have adequately responded to my previous concerns and I hope the paper is well utilized within the community. 

Author Response

Thank you very much for having our paper entitled " An Improved Model Based Detection of Urban Impervious Surfaces Using Multiple Features Extracted from ROSIS-3 Hyperspectral Images" reviewed again and sending us a minor comment regarding language use issue. We have carefully revised the manuscript to improve English language description. Additionally, we have checked the entire sections of the manuscript including main text, figures, tables, equations and references to ensure its compliance with the style or format of Remote Sensing. All modifications are marked in red ink in the revised version. 

Thank you and best regards.

Yours sincerely,

Yuliang Wang, Huiyi Su, Mingshi Li